# Visible colorimetric dosimetry of UV and ionizing radiations by a dual-module photochromic nanocluster

Huangjie Lu[1,2,3,6], Jian Xie[4,6], Xin-Yu Wang[2], Yaxing Wang[4], Zi-Jian Li [1,3], Kariem Diefenbach[1], Qing-Jiang Pan [2✉], Yuan Qian[1,3], Jian-Qiang Wang [1,3,5], Shuao Wang [4✉] & Jian Lin [1,3✉]

Radiation dosimeters displaying conspicuous response of irradiance are highly desirable, owing to the growing demand of monitoring high-energy radiation and environmental exposure. Herein, we present a case of dosimetry based on a discrete nanocluster, $[Th_6(OH)_4(O)_4(H_2O)_6](TPC)_8(HCOO)_4\cdot4DMF\cdot H_2O$ (Th-SINAP-100), by judiciously incorporating heavy $Th_6$ polynuclear centers as radiation attenuator and organic linkers as photoresponsive sensor. Interestingly, dual-module photochromic transitions upon multiple external stimuli including UV, β-ray, and γ-ray are integrated into this single material. The striking color change, and more significantly, the visible color transition of luminescence in response to accumulating radiation dose allow an on-site quantitative platform for naked-eye detection of ionization radiations over a broad range (1–80 kGy). Single crystal X-ray diffraction and density functional theory calculations reveal that the dual-module photochromism can be attributed to the $\pi(TPC) \rightarrow \pi^\star(TPC)$ intermolecular charge transfer driven by enhanced π-π stacking interaction between the adjacent TPC moieties upon irradiation.

[1] Key Laboratory of Interfacial Physics and Technology, Shanghai Institute of Applied Physics, Chinese Academy of Sciences, Shanghai, China. [2] Key Laboratory of Functional Inorganic Material Chemistry (Ministry of Education), School of Chemistry and Materials Science, Heilongjiang University, Harbin, China. [3] University of Chinese Academy of Sciences, Beijing, China. [4] School for Radiological and Interdisciplinary Sciences (RAD-X) and Collaborative Innovation Center of Radiation Medicine of Jiangsu Higher Education Institutions, Soochow University, Suzhou, China. [5] Dalian National Laboratory for Clean Energy, Dalian, China. [6] These authors contributed equally: Huangjie Lu, Jian Xie. ✉email: panqjitc@163.com; shuaowang@suda.edu.cn; linjian@sinap.ac.cn

D eveloping radiation dosimeters has been a very active research area of inquiry for the past decades, owning to the cumulative demands of such materials in multifarious fields, including nuclear industry, medical radiation, cosmic exploration, and food irradiation[1–6]. Quantitatively detection of a radiation dose usually occurs by converting incident energy to a low-energy photon response, electronic current, or threshold voltage by radiation detectors[7]. These types of dosimeters, e.g., ionization chambers, thermoluminescent dosimeters, and semiconductors, etc., require readout instruments such as a photomultiplier and electrometer, or even spectrophotometer to obtain quantitative information[8–12]. On the other hand, radiochromic films allow for direct readout and onsite assessment of radiation exposure via visible color change[13]. However, only qualitative or semi-quantitative characterization of radiation can be accomplished without the assistance of a spectrophotometer[14]. Therefore, the quest for developing radiation dosimeters that provide a more straightforward route for dosimetry of ionizing radiations remains ongoing.

Metal-organic complexes featuring photoinduced changes of electronic properties, photochromism and photoluminescence (PL), represent a promising dosimetry platform for colorimetric visualization of ionizing radiation[15–17]. The incorporation of a metal center with a high atomic number (Z) endows the material with increased radiation attenuation efficiency[18,19]. Furthermore, coordinating linkers decorated with a photoactive ligand, such as pyridine, viologen, and azobenzene, allows for charge transfer and formation of radical species upon irradiation, inducing visualizable color transitions[20–24]. Metal-organic complexes featuring ionizing radiation induced photochromism remain to be a small group among the numerous numbers of photochromic materials; systems showing photochromically modulated luminescence are even less common[25–27]. In addition, the PL modulations of these materials are limited to simple on/off switching with coloring/bleaching behaviors, while the emission colors do not change[2,28].

Herein, we present a dual-module photochromic nanocluster, $[Th_6(OH)_4(O)_4(H_2O)_6](TPC)_8(HCOO)_4·4DMF·H_2O$ (Th-SINAP-100, TPC = 2,2′:6′,2″-terpyridine-4′-carboxylate), featuring both color and luminescence photochromism within a single material upon UV, β, or γ-ray irradiation. The strong Lewis acidity of $Th^{IV}$ promotes the formation of a dense hexanuclear $[Th_6(OH)_4(O)_4(H_2O)_6]^{12+}$ cluster via hydrolysis and nucleation, which together with its third largest atomic number value among all naturally occurring elements, ensure large photoelectric attenuation coefficiency of affording material[29]. The photochromic transition from purple to yellow facilitates qualitative detection of ionization radiation. More significantly, the stepwise dual-emission PL modulation upon irradiation dose from blue, to cyan, and to green of Th-SINAP-100 can ultimately serve as a visible colorimetric dosimeter, like the widely used pH strip, for radiation dosimetry. Moreover, the radiation dose can be more accurately correlated with the RGB parameters of PL collected simply using a digital camera and graphic software.

## Results

**Synthesis and molecular structure.** Solvothermal reaction of thorium nitrate, 2,2′:6′,2″-terpyridine-4′-carboxylic acid (HTPC), and HCl in DMF affords purple crystals of Th-SINAP-100. Single crystal X-ray diffraction (SCXRD) reveals that Th-SINAP-100 crystallizes in a triclinic space group $P\bar{1}$ (Supplementary Table 1) and it features a 0D discrete nanocluster topology. The asymmetric unit consists of one $[Th_6(OH)_4(O)_4(H_2O)_6]^{12+}$ hexamer cluster decorated by eight TPC ligands and four formate anions (Fig. 1a). The $[Th_6(OH)_4O_4(H_2O)_6]^{12+}$ core is composed of six

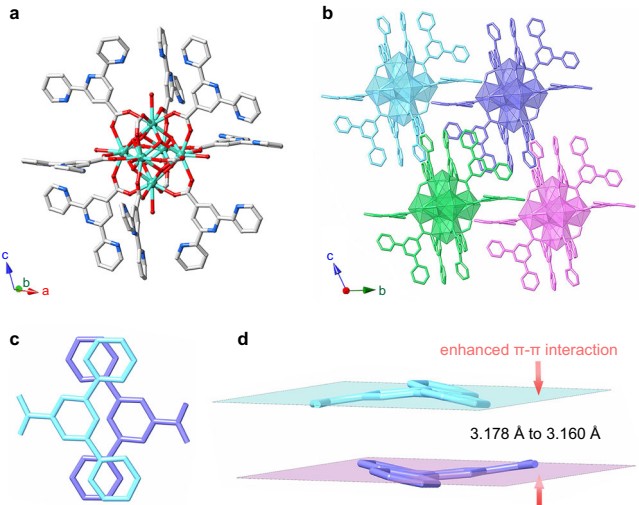

**Fig. 1 Crystal structure of Th-SINAP-100. a** Molecular structure of Th-SINAP-100. Color code: Th green, O red, N blue, and C gray. H atoms are omitted for clarity. **b** Topological representation of the packing of 0D clusters. **c** Top view showing the π–π interactions in the structure of Th-SINAP-100. **d** Side view showing the π–π interactions in the structure of Th-SINAP-100.

$Th^{IV}$ ions bridged by four $\mu_3$-O and four $\mu_3$-OH groups with six capping $H_2O$ molecules located on each vertex of the octahedron, which is similar to other hexameric actinide cluster species (Supplementary Fig. 1)[30,31]. The $Th^{IV}$ cation adopts a capped square antiprismatic geometry, within which four O atoms are donated from $\mu^3$-O/OH groups, four from TPC or formate ligands, and one from a $H_2O$ molecule. It is noteworthy that the Zr, Hf, and Ce analogues of Th-SINAP-100 cannot be obtained under comparable conditions since cations in $[M_6(OH)_4O_4]^{12+}$ (M = Zr, Hf, and Ce) clusters are typically eight-coordinated without being decorated with water molecules[32,33]. The 0D clusters are further packed via weak intermolecular van der Waals interaction and aromatic π-π stacking interactions as shown in Fig. 1b. Two neighboring TPC ligands are approximately parallel with a nearest interlamellar distance of 3.178 Å, suggesting the presence of π–π interactions between the moieties (Fig. 1c, d). The phase purity of bulk samples of Th-SINAP-100 was confirmed by powder X-ray diffraction (PXRD) (Supplementary Fig. 2).

**Dual-module photochromism.** Upon UV (365 nm), β-ray (1.2 MeV), or γ-ray ($2.22 \times 10^{15}$ Bq) irradiation, Th-SINAP-100 exhibits photochromic transformation from purple to yellow as shown in Fig. 2a. While UV and X-ray induced photochromic switching behavior is relatively common, γ-ray responsive photochromism has been barely reported[34,35] and β-ray induced photochromism has remained elusive. The evolution of UV-Vis adsorption spectra was initially acquired under exposure to 365 nm UV light, as depicted in Fig. 2b. The purple crystal displayed three adsorption maxima at approximately 340, 410, and 580 nm. The bands at 340 nm diminished gradually as a function of irradiation dose, while the intensities of the peaks at 410 and 580 nm increased and maximized after 0.04 mJ UV radiation (Fig. 2b). Upon further irradiation, two peaks at short wavelength area (340 and 410 nm) slowly merged into a broad band and the one at 580 nm was diminished, which can be considered as the second stage (Fig. 2b, inset). Similar spectral

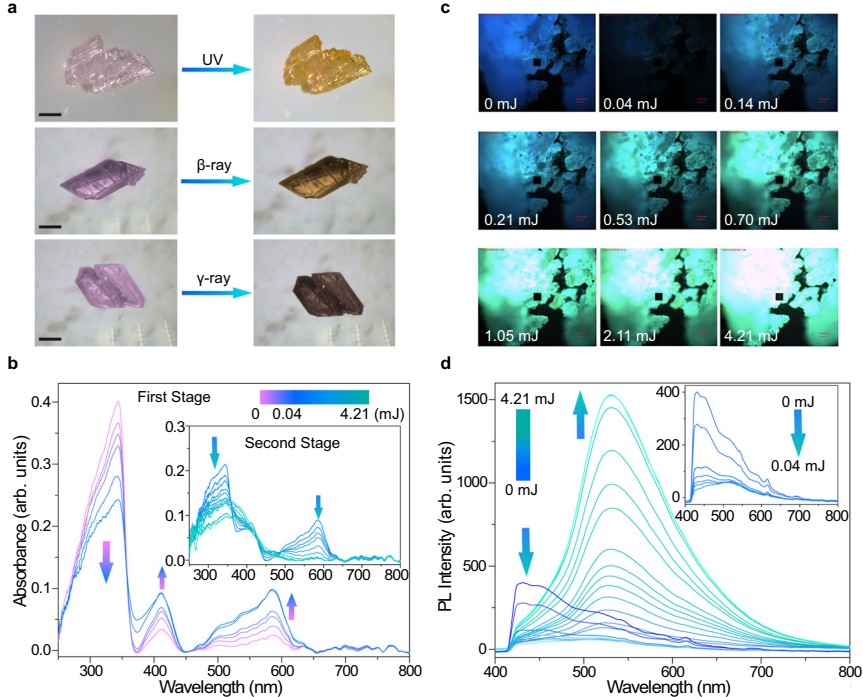

**Fig. 2 Dual-module photochromism of Th-SINAP-100. a** Photochromism of Th-SINAP-100 upon UV-, β-ray, and γ-ray irradiation. The black scale bars in photographs represent 200 μm in length. **b** Evolution of the solid-state UV-Vis absorption spectra of Th-SINAP-100 crystal after exposure to 0–0.04 mJ UV irradiation doses ($\lambda_{ex}$ = 365 nm, 2 mW) at 25 °C. Inset: solid-state UV-Vis absorption spectra of Th-SINAP-100 crystal after exposure to 0.04–4.21 mJ UV irradiation dose. The UV-Vis spectra were collected on a crystal of Th-SINAP-100 from 250 to 800 nm based on the average intensities of 50 scans with a scan time of 2 ms. **c** Snapshots of bulk Th-SINAP-100 crystals upon 365 nm UV irradiation. **d** Evolution of the solid-state photoluminescence (PL) spectra of Th-SINAP-100 crystal after exposure to 0–4.21 mJ UV irradiation doses ($\lambda_{ex}$ = 365 nm, 2 mW) at 25 °C. Inset: PL spectra of Th-SINAP-100 after exposure to 0–0.04 mJ UV irradiation doses. The PL spectra were collected on a crystal of Th-SINAP-100 from 400 to 800 nm based on the average intensities of 5 scans with a scan time of 500 ms.

evolutions upon β- and γ-ray irradiation were observed for Th-SINAP-100 as shown in Supplementary Fig. 3. One should note that the TPC ligand did not show obvious photoinduced color change even after 38.09 mJ UV radiation, which is consistent with its negligible variation of UV-Vis spectra upon irradiation (Supplementary Fig. 4).

More interestingly, Th-SINAP-100 exhibits a luminescence photochromic transition with PL color evolution upon continuous UV, β-ray, and γ-ray irradiations (Fig. 2c and Supplementary Fig. 5). Such PL modulations upon exposure to external stimuli have only been documented in [((PyrO)₃tacn)Gd(THF)] and Pb(2-MTA)DMF (SCU-200)[18,36]. Moreover, single material featuring dual-module color and luminescence photochromism is uncommon. Explicitly, nonirradiated Th-SINAP-100 displayed blue emission under excitation with 365 nm light, corresponding to the broad PL maxima at 432 nm in its PL spectrum (Fig. 2d, inset). The intensity of $PL_{432}$ drastically decreased as a function of exposure time and approximately 8% of the original intensity preserved upon 0.04 mJ UV irradiation. This photo bleaching process can be correlated with the first stage of photochromism, wherein the absorption at 340 nm is suppressed. Further UV irradiation resulted in the emergence of a broad PL band centered at 532 nm and this change can be seen by gradate transitions of the emission color from blue, to cyan, and eventually to green (Fig. 2d). The intensity of the $PL_{532}$ band is saturated after 4.21 mJ UV exposure and is significantly higher than that of the initial blue emission band. Similar phenomenon was observed when Th-SINAP-100 was exposed to increasing doses of β- or γ-ray irradiation as shown in Supplementary Fig. 5. Importantly, Th-SINAP-100 and TPC ligand exhibit different luminescence

features, within which the later one experiences a decoloration process (Supplementary Fig. 6).

**Fluorometric dosimeter.** The coloration information can be stored and the PL intensity of Th-SINAP-100 remained unchanged for at least 48 h when kept in dark under ambient condition, implying the potential use of Th-SINAP-100 as a radiation dosimeter (Supplementary Fig. 7). The UV radiation dosage can be simply estimated by the PL color from the observation by the naked eyes and more accurately correlated with the CIE chromaticity coordinate as a numerical representation (Fig. 3a and Supplementary Table 2). Alternatively, the ratiometric dual-emission with opposite PL dependence on dose could be utilized for detection of radiation with higher accuracy. This type of spectroscopic character has been employed for the efficient sensitization of cation species, but was barely applied for quantification of ionization radiation[37]. The inclusion of an addition emission band as a reference not only allows fine-tuning the energy transfer via photo-activation, but also eliminates possible interference, e.g., drift of the light resource and detector, to improve detection accuracy[37]. The declined intensity of PL at 432 nm and inclined emission band at 532 nm contribute to the continuous increase of $I_{532}/I_{432}$ ratio as a function of irradiation dose. Linear correlations with three distinct slopes between the PL intensity ratios of $I_{532}/I_{432}$ and the UV doses can be established as shown in Fig. 3b, leading to an accurate UV detection over a broad dose ranging from 0 to 4.2 mJ. The sharper transition of $I_{532}/I_{432}$, at a low dose region compared to those of $I_{532}$ and $I_{432}$, suggests more accurate dose detection and less interference. To

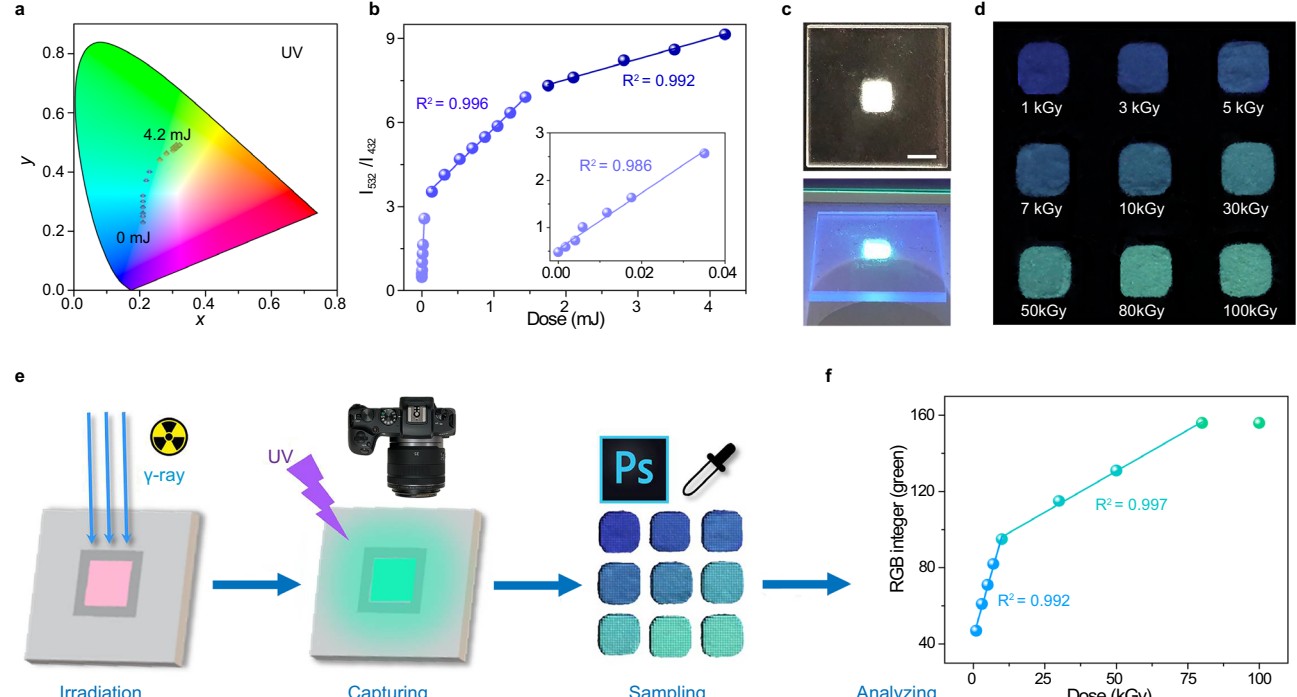

**Fig. 3 Fluorometric and colorimetric dosimeter. a** CIE chromaticity coordinates of Th-SINAP-100 as a function of UV irradiation dose. **b** Linear fittings of the intensity ratio $I_{532}/I_{432}$ as a function of UV dosage. The inset is the linear fitting in the low dose range from 0 to 0.04 mJ. **c** Photos of Th-SINAP-100-based dosimeter. The white scale bar in photograph represents 5 mm in length. **d** Color matching chart prepared by irradiated Th-SINAP-100 with different γ-ray dose. **e** A schematic illustration of colorimetric quantification method based on RGB scale. **f** Linear fittings of the RGB integer (green) of Th-SINAP-100 as a function of γ-ray dose.

facilitate the real-world application, the Th-SINAP-100 poly-crystalline can be further fabricated with polyvinylidene fluoride (PVDF) polymer into a flexible strip (Supplementary Fig. 8a)[8]. As shown in Supplementary Fig. 8b, the color of strip is initially purple under excitation with 254 nm UV light. Continuous UV irradiation (350 mA, 25 W) results in gradual transitions of emission colors from purple, to blue, to cyan, and to green, implying the potential applicability of Th-SINAP-100@PVDF strip in radiation dosimetry.

**Direct readout colorimetric dosimeter.** Promoted by the step-wise luminescence photochromism of Th-SINAP-100 from blue, to cyan, and to green upon γ-ray radiation, a colorimetric dosimeter for visible quantification of irradiation dose without a separate readout apparatus was developed. Th-SINAP-100 was loaded on a customized quartz slide as a carry-on dosimeter with a dimension of $2 \times 2 \, cm^2$ (Fig. 3c). A color matching chart was prepared by exposing Th-SINAP-100 to γ-ray irradiations with precisely controlled doses and can be utilized as a reference for unknown doses. As shown in Fig. 3d, Th-SINAP-100 exhibits eight distinct PL colors in response to 1, 3, 5, 7, 10, 30, 50, and 80 kGy γ-ray irradiation, while the color remains unchanged when the dose is further increased to 100 kGy. The upper limit of radiation dose is approximately 80 kGy, which is higher than currently existing radio-photoluminescence dosimeters, e.g., Pb (2-MTA)·DMF (SCU-200) and LiF:Mg (TLD-100)[18,38]. The dose range of Th-SINAP-100 is comparable to commercialized radiochromic film dosimeter GEX B3 and among one the highest of all dosimeters[13,39]. This feature makes Th-SINAP-100 parti-cularly suitable for high dose (kGy level) monitoring in diverse fields including synchrotron radiation monitoring, food irradia-tion, medical sterilization, and radiodegradation[40,41]. The operation ranges of metal-ion-doped inorganic dosimeters, e.g.,

Ag-doped phosphate glass and $Mg^{2+}$-doped LiF (LiF:Mg), are saturated at relatively low dose due to their limited number of PL metal centers[42]. In contrast, the larger set of radiation sensitive organic linkers in metal-organic materials allows for a higher saturation point and a wider operation range. Furthermore, Th-SINAP-100 offers substantial advantages over traditional radia-tion dosimeters, e.g., film badge, quartz fiber, Geiger tube, metal–oxide–semiconductor field-effect transistor (MOSFET) dosimeter, and TLD, in terms of onsite analysis, direct readout, user-friendly, low cost, or portability.

More gratifyingly, the RGB components can be extracted from the PL by simply capturing and analyzing the optical image using a digital camera and graphic software, respectively (Fig. 3e). The green (G) component of the three primary colors (RGB) as a function of γ-ray irradiation dose was plotted as shown in Fig. 3f. Two regions of good linear responses with $R^2$ values of 0.992 and 0.997 were established in the ranges of 1–10 kGy and 10–80 kGy, respectively. This quantitative colorimetric analysis based on a RGB scale leads to more accurate radiation dosimetry over such ranges.

**Radiolytic and chemical stabilities.** The radiolytic stability of Th-SINAP-100 was examined by exposing polycrystalline Th-SINAP-100 under β and γ-rays with dose rates of 150 kGy h$^{-1}$ and 11.8 kGy h$^{-1}$, respectively. The structure of Th-SINAP-100 remained unchanged even after 1 MGy dose of both β and γ-ray irradiation, as evidenced by their corresponding PXRD patterns and FTIR spectra (Fig. 4 and Supplementary Fig. 9). Metal-organic frameworks (MOFs) featuring high radiation-resistance were known as exemplified by $Th_6O_4(OH)_4(H_2O)_6(C_{22}O_6H_{12})_6$ (TOF-16) (Supplementary Table 3)[43]. The investigation on radiation stability of cluster species has not been conducted and the only relevant case is an ultrastable $[Hf_{13}]$ nanocluster, which

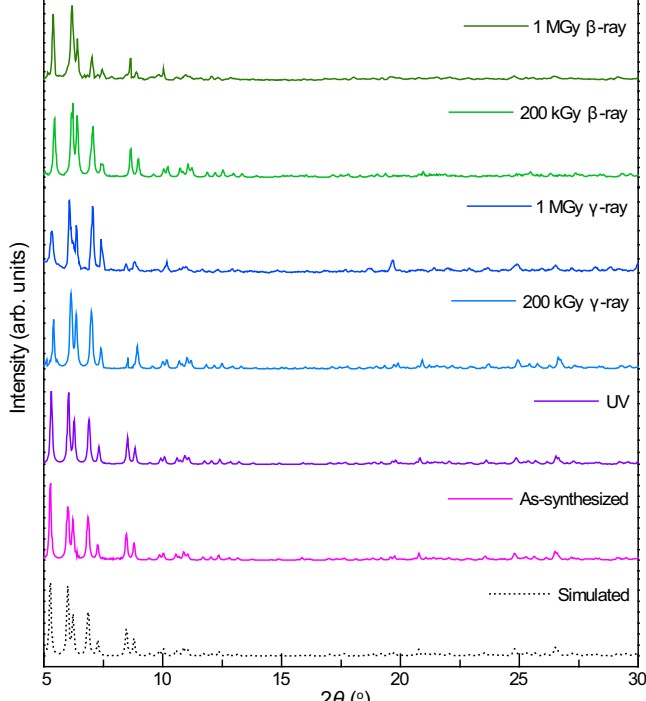

**Fig. 4 Radiolytic stability of Th-SINAP-100.** PXRD patterns of Th-SINAP-100 before and after irradiation with UV, β-ray, or γ-ray.

is highly resistant to both concentrated acid and alkali[44,45]. Th-SINAP-100 represents a radiation-resistant polynuclear cluster and one of the highly radiolytic metal-organic materials in general. Furthermore, PXRD studies indicate that the structure of Th-SINAP-100 remains intact at relative humidity (RH) of 35, 55, 75, and 95% in air (Supplementary Fig. 10a). In addition, Th-SINAP-100 displays excellent hydrolytic stabilities in aqueous solutions with pH values ranging from 4 to 8 (Supplementary Fig. 10b). These radiolytic and chemical stabilities could be attributed to the conjugated π-electrons of terpyridine to stabilize the organic linkers and the affording cluster[46]. The π–π interactions between the neighboring terpyridine moieties further stabilize the metal clusters. Moreover, PXRD and thermogravimetric analysis (TGA) demonstrated that Th-SINAP-100 exhibits decent thermal stability and is stable up to 150 °C (Supplementary Figs. 10c and 11).

**Mechanism of photochromism and photoluminescence modulation.** The photochromism can be explained by a variety of mechanisms, including phase transitions[47], generation of radical species[48], electron-transfer (redox) chemical process[49], tautomeric shift[50], photoinduced charge transfer, etc[51–54]. To gain insight into the photocoloration mechanism of Th-SINAP-100, the PXRD patterns and FTIR spectra before and after irradiation were collected, showing no obvious variations of structure and chemical constituent (Fig. 4 and Supplementary Fig. 9). The recorded electron paramagnetic resonance (EPR) spectra indicate that Th-SINAP-100 is EPR silent before irradiation but shows a sharp resonance after UV-, β-ray, and γ-ray irradiation at $g = 2.0044$, corresponding to the tensor value ($g = 2.0023$) of a free electron (Fig. 5a)[25]. The PL intensity of Th-SINAP-100 remains approximately unchanged after removal of continuous radiation source, suggesting that the affording radicals are stable under ambient conditions (Supplementary Fig. 7). These observations led to the tentative speculation that the photochromism and PL modulation of Th-SINAP-100 originate from the generation of

radical species. However, neither photochromism nor similar PL modulation was observed for HTPC despite the fact that radical species ($g = 1.9166$ in EPR) can be identified (Supplementary Fig. 12), implying an alternative mechanism to elucidate these photoresponsive properties.

To further elucidate the photochromic behavior of Th-SINAP-100, we resort to density functional theory (DFT) calculations. Experimentally synthesized Th-SINAP-100 (abbreviated as TS) and two groups of simplified complex models, $[Th_6(O-H)_4(O)_4(H_2O)_6](TPC)$ (TS-m) and $[Th_6(OH)_4(O)_4(H_2O)_6]$ $(TPC)_2$ (TS-d), were fully optimized (Fig. 5b). TS-m1 and TS-d1 were found to be the most energetically stable states. The computed Th−O distances exhibit the order of $Th-O_{oxo/hydroxo} < Th-O_{TPC} \approx Th-O_{HCOO} < Th-O_W$, whose values agree well with the experimental ones (Supplementary Tables 4 and 5). Furthermore, their bond orders range from 0.34 to 0.80, indicating weak single bond character between $Th^{IV}$ and oxo atoms (Supplementary Table 6).

Time-dependent DFT calculations (TD-DFT) were carried out by using the most stable TS-d1 model. The experimental intense absorption at 340 nm is comparable to the theoretically simulated one at 360 nm (Supplementary Table 7). Analysis of excited-state wave function unraveled that the band arises from the π(TPC) → π*(TPC) intramolecular charge transfer (IaMCT) mixed with a partial contribution of the thorium-oxo moiety. It is intuitively supported by the orbitals involved in the 364 nm transition with the largest oscillator strength (Fig. 5c). Additionally, a relatively low-energy transition was calculated at 392 nm and its virtually transition-forbidden nature implies very slight contribution to the experimental 410 nm peak (Supplementary Fig. 13). Given the limitation of single-cluster models, the calculations of TS-d1 and TS-m1 did not yield absorptions that could be comparable to experimental peaks at longer wavelengths (410 and 580 nm) (Fig. 2b).

To gain a deeper insight into the mechanism of photocoloration, detailed structural comparisons were performed on a single crystal of Th-SINAP-100 before and after the γ-ray irradiated with an accumulated dose of 200 kGy. Both datasets were collected under identical conditions including temperature, measurement angles, and exposure time. SCXRD studies revealed that the overall topologies before and after irradiation remain unchanged, whereas the unit cell parameters $a$, $b$, $c$, and $V$ shrink from 14.526(1) Å, 17.246(2) Å, 18.504(2) Å, and 3908.7(6) Å$^3$ to 14.500(6) Å, 17.173(7) Å, 18.368(9) Å, and 3875.0(3) Å$^3$, respectively, implying a more compact molecular packing after irradiation (Supplementary Table 1)[55]. Concomitantly, the interplane distances of the π–π stacking of the terpyridyl rings along the $c$ axis decrease from 3.178 to 3.160 Å (Fig. 1d and Supplementary Table 4). These π–π interactions are enhanced allowing $π → π*$ transition to occur between adjacent TPC ligands, which enables intermolecular charge transfer (IeMCT) and eventually gives rise to the absorption bands at 410 and 580 nm for Th-SINAP-100.

In brief, the high-energy experimental band (340 nm) is attributable to IaMCT in nature, while the two lower-energy ones (410 and 580 nm) are primarily of IeMCT. More detailed results from TD-DFT calculations show that the absorption band at 410 nm features a negligible IaMCT mixture. Moreover, TPC precursor does not exhibit photochromism owing to the lack of such π-π stacking interactions. The luminescence photochromism of Th-SINAP-100 was further rationalized by DFT calculations. The assembly of $Th^{IV}$ and TPC ligands into the molecular complex facilitates the π–π stacking of TPC moieties in the crystalline environment. Consequently, two types of excited states, TS* originating from IaMCT and (TS-TS)* from IeMCT, can be created under excitation. Since being stabilized by the

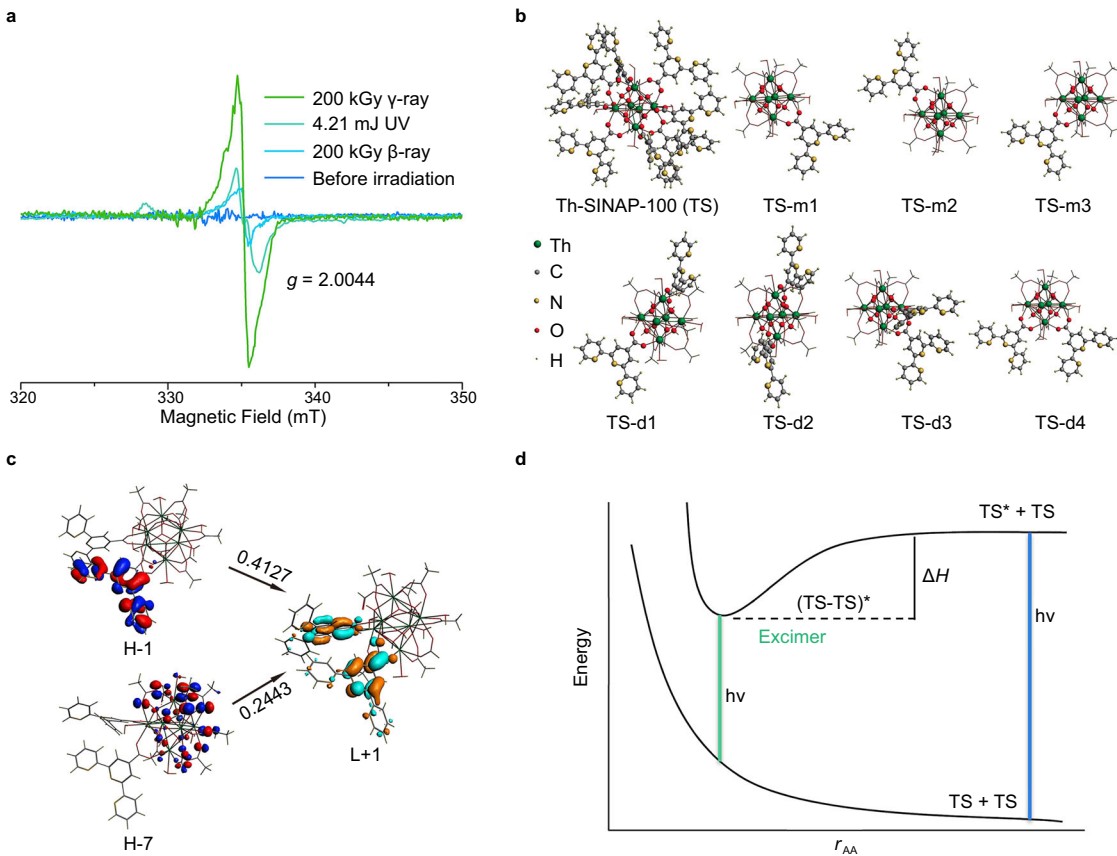

**Fig. 5 Mechanism of photochromism and photoluminescence modulation. a** EPR spectra of Th-SINAP-100 before and after irradiation. **b** Optimized structures of the experimentally synthesized complex and selected simplified models of Th-SINAP-100 (TS). **c** Orbital diagrams of the 364 nm electron transitions for TS-d1 from the TD-DFT calculations. **d** The formation of excimer (TS-TS)* that is stabilized by charge transfer, along with the light emitted by (TS−TS)* and TS* + TS.

charge transfer, the excimer (TS-TS)* is lower in energy than the discrete TS* + TS, corresponding to an emission band at longer wavelength region (532 nm vs. 432 nm) (Fig. 5d).

## Discussion

We have developed a case of a dual-module photochromic metal-organic nanocluster, Th-SINAP-100, that delivers direct readout colorimetric dosimetry for UV and ionizing radiations. The combination of dense $[Th_6(OH)_4(O)_4(H_2O)_6]^{12+}$ cores as radiation attenuators and TPC ligands as photoresponsive sensors gives rise to both color and luminescence photochromic transitions incorporated into a single material. Remarkably, Th-SINAP-100 displays β-ray induced photochromism. The visible response of PL allows Th-SINAP-100 to function as a colorimetric dosimeter for ionizing radiations, like the widely used pH stripe, overcoming the reliance on readout instruments for radiation dosimetry. Thanks to the high density of radiation sensitive organic linkers in its structure, a record high dose (1 MGy) of dosimetry compared to existing radio-photoluminescence materials is achieved. SCXRD studies and DFT/TD-DFT calculations reveal that the enhanced π–π stacking interaction between the adjacent TPC moieties upon irradiation induce the IaMCT→IeMCT conversion, ultimately leading to the photoinduced peculiarities.

## Methods

Caution! Th-232 and its daughter nucleus Ra-228 are both radioactive. All of the thorium compounds used and investigated were operated in an authorized laboratory designed for actinide element studies. Standard protections for radioactive materials should be followed.

**Materials**. $Th(NO_3)_4 \cdot 6H_2O$ (99.9%, Changchun Institute of Applied Chemistry, Chinese Academy of Sciences), HTPC (99%, Jilin Chinese Academy of Sciences—Yanshen Technology Co., Ltd), DMF (99.5%, Aladdin), and concentrated HCl (AR, 36~38%, Sinopharm Chemistry Reagent Co., Ltd) were used as received from commercial suppliers without further purification.

**Synthesis**. $Th(NO_3)_4 \cdot 6H_2O$ (29.4 mg, 0.05 mmol), HTPC (13.8 mg, 0.05 mmol), concentrated HCl (100 μL), DMF (1 mL), and deionized $H_2O$ (1 mL) were loaded into a 5 mL glass vial and heated in an oven at 100 °C for 24 h. After cooling to room, purple block crystals of Th-SINAP-100 were isolated, washed with deionized water, and dried under ambient conditions. The yield of Th-SINAP-100 was calculated to be 49(1)% based on HTPC.

**Crystallographic analysis**. Single crystal X-ray diffraction data of Th-SINAP-100 were collected on a Bruker D8-Venture diffractometer equipped with an IμS 3.0 Mo Kα X-ray source ($\lambda = 0.71073$ Å) and a Photon100 CMOS detector at 298 K. SCXRD study revealed the presence of large electron density within the void space of Th-SINAP-100. The solvent species were excluded in the crystal structure using SQUEEZE routine of PLATON and the electron counts per cell for Th-SINAP-100 were found to be 309 $e^-$ [56]. Powder X-ray diffraction (PXRD) patterns were collected on a Bruker D8 Advance diffractometer equipped with a Cu Kα radiation ($\lambda = 1.54056$ Å) and a Lynxeye detector.

**Physical property measurements**. The solid-state UV-Vis absorption and luminescence spectra of Th-SINAP-100 were collected on a Craic Technologies microspectrophotometer. The electron paramagnetic resonance data were recorded on a JEOL-FA200 spectrometer at the X-band with 100 kHz field modulation at room temperature. The Fourier transform infrared spectroscopy study was carried out a Thermo Nicolet 6700 spectrometer equipped with a diamond attenuated total reflectance (ATR) accessory. Elemental analyses of C, H, and N were performed on Th-SINAP-100 with a Vario EL Cube elemental analyser. The weight contents of N, C, and H are 9.04%, 39.77%, and 3.775%, respectively, suggesting the presence of DMF and $H_2O$ as solvent species in Th-SINAP-100. The number of DMF molecules in the void was calculated to be four per molecular formula. Thermogravimetric analysis was performed on a NETZSCH STA 449 F3 Jupiter thermal

analyzer at a heating rate of 10 °C min$^{-1}$ under a nitrogen flow, showing an initial solvent loss of 7.24 w%, which can be attributed to the departure of four DMF and one $H_2O$ molecules (*calcd.* 7.16 w%) (Supplementary Fig. 11).

**Radiation-resistance measurements.** The radiation-resistance of Th-SINAP-100 was inspected by irradiating polycrystalline samples with β-ray or γ-ray. The sources of β-ray and γ-ray were provided by an electron accelerator (1.2 MeV) and a $^{60}Co$ irradiation source ($2.22 \times 10^{15}$ Bq), respectively. Th-SINAP-100 was irradiated with two different doses (200 kGy and 1 MGy) with dose rates of 150 and 11.8 kGy h$^{-1}$ for β-ray and γ-ray, respectively. PXRD and FTIR analyses on the irradiated samples were performed to confirm the radiation-resistance of Th-SINAP-100.

**Computational methods.** Density functional theory (DFT) calculations were performed to optimize the structure of the experimentally obtained Th-SINAP-100 (abbreviated as TS). The inclusion of 300 atoms and 2278 electrons within this large molecular complex results in high computational demand to elucidate the electronic structure, particularly excited states, of Th-SINAP-100. Therefore, the complex was simplified by partially replacing bulky TPC ligand with $CH_3COO^-$ moiety as shown in Fig. 5b. Consequently, two simplified complex modes, TS monomer (TS-m) and dimer (TS-d), which feature one and two TPC ligands in the molecule, respectively, were selected. The possible isomers were optimized for both modes and the structures of the most energetically stable isomers are depicted in Fig. 5b.

In geometry optimizations, the PBE functional of the generalized gradient approximation (GGA) was applied. A scalar relativistic four-component all-electron approach and all-electron correlation-consistent double-ζ polarized quality basis sets were used[57]. Subsequent analytical frequency calculations were used to confirm the nature of the stationary points on the potential energy surface[57]. The zero-point vibrational energy (ZPVE) and entropy were further obtained. Population-based Mayer[3] bond orders and Mulliken charges were calculated. All these calculations were accomplished with the Priroda code[57,58].

The electronic structures were calculated with the ADF 2014 code while employing the above optimized TS-m1 and TS-d1 geometries[59–61]. An integration parameter of 6.0 was applied. The ZORA scalar relativistic approach of van Lenthe et al. was employed, associated with the Slater-type TZP basis sets[62–65]. The time-dependent density functional theory (TD-DFT) with the PBE functional was performed to calculate electronic absorption spectra (Supplementary Fig. 14). Fifty spin-allowed excited states were calculated for model complexes, and eighty ones for the TPC ligand. Considering the effect of the environmental media on above properties, we used the Conductor-Like Screening Model (COSMO)[66,67] to simulate the crystalline environment surrounding the complex. The solvent water was utilized with the dielectric constant of 78.38. Klamt radii were used for the main group atoms (H = 1.30 Å, C = 2.00 Å, O = 1.72 Å, and N = 1.83 Å) and for the thorium atom (2.00 Å)[68–70].

## Data availability

The data that support the findings of this paper are available in the paper and supplementary information files. The X-ray crystallographic data for the structure reported in this article has been deposited at the Cambridge Crystallographic Data Centre (CCDC) with the number of CCDC 2050840-2050841. These data can be obtained free of charge from The Cambridge Crystallographic Data Centre via www.ccdc.cam.ac.uk/data_request/cif. Any further relevant data are available from the authors upon reasonable request. Source data are provided with this paper.

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

## Acknowledgements

We cordially thank Prof. Ning Chen (Chemical Engineering and Materials Science, Soochow University) and Prof. Zhengyang Zhou (Shanghai Institute of Ceramics, Chinese Academy of Sciences) for helpful discussions. This work was supported by the National Natural Science Foundation of China (22076196, 21876182, 21825601, 21701184, and 21790374), the Strategic Priority Research Program of the Chinese Academy of Sciences (XDA21000000), and the K.C.Wong Education Foundation (GJTD-2018-10).

## Author contributions

J.L. conceived and supervised the project. H.L designed and synthesized the material. H.L., J.X., Y.W., and Y.Q. carried out the characterizations and analyzed the data. Z.J.L. aided in the irradiation experiments. X.Y.W. and Q.J.P. carried out the theoretical calculation. H.L., K.D., Q.J.P., J.Q.W., S.W., and J.L. wrote the paper. All authors discussed the results and commented on the manuscript.

## Competing interests

A patent "Metal-organic hybrid material for ionization radiation dosimetry, 202010501386X." on the related content has been filed by J.-Q.W., H.L., J.L., and Shanghai Institute of Applied Physics, Chinese Academy of Sciences. The remaining authors declare no competing interests.
