## [Peer Review File · Nature Communications]

REVIEWER COMMENTS

Reviewer #1 (Remarks to the Author):

The paper by Lin and co-workers provides an impressive demonstration of a dual-module photochromic metal-organic nanocluster, Th-SINAP-100, as a key element for colorimetric dosimetry for UV and ionizing radiations. The judicious synergy between radiation attenuators and photo responsive ligands is responsible for the rare color and PL simultaneously demonstrated in a single material.

While the methodology is well described and the results are very surprising especially taking into account the uncommon β -ray induced photochromic effect here observed, some aspects should be taken into account if continuous dose monitoring applications are envisaged here.

The influence of the environmental conditions on the observed results should be assessed especially in terms of relative humidity and temperature

Moreover, did the authors observe any recovery effect upon irradiation as function of time? At which the extent the observed properties could be maintained at nanoscale, i.e how the radiation induced sensitivity could depend on the thickness on the sensitive material?

The weak response observed a low dose rate very important for most of the relevant clinical applications is a shortcoming, could the author comment on potential degree of freedom to further extend the measuring of this intriguing nanocluster

Finally, demonstration of encapsulation of the sensitive nanocluster in simple shapes for envisioning potential applications (fibers, strips ...) is strongly encouraged.

Reviewer #2 (Remarks to the Author):

The evaluation of the crystallographic data reveals that the authors have alerts A in their cif check pdf. MOFs are porous structures, and therefore, it is expected to observe large voids. However, there are several methods to take them into account in the crystallographic models. Unfortunately, the authors do not address this problem properly. One way to treat the problem is to find and model the disordered solvent molecules which are located inside MOF pores. Instead, the authors have assigned the residual electron density to oxygen atoms which are randomly distributed in the voids of MOFs. Another way to treat this issue is to use the SQUEEZE procedure (freely available). However, in the case, the authors should count number of electrons and estimate the correct number of the solvent molecules inside a framework. Both these methods are commonly applied to the MOF structures.

To conclude, application of any of these methods is necessary for publication of the collected crystallographic data.

Reviewer #3 (Remarks to the Author):

This manuscript presents a thorium nanocluster that is composed of $\text{Th}_6\text{O}_4(\text{OH})_4$ cores decorated by photosensitive terpyridine derivatives. Interestingly, the as-reported compound Th-SINAP-100 exhibited a rather unusual dual-module photochromism upon multiple types of irradiations, including UV light, β -ray, and γ -ray. This feature allows visual broad dosimetry of ionization radiations over a range from 1 to 80 kGy. The mechanism of dual-module photochromism was thoroughly studied by SCXRD and DFT calculations. This new

colorimetric dosimeter based on nanocluster structure should be of great interest to a wide range of readers, including chemists and material scientists. I recommend this work to be accepted after addressing the following issues.

1. Molecular formula and compound name should be mentioned in the abstract.
2. In Fig. 2a, rules in all images should be added.
3. The author mentioned "The PL intensity of Th-SINAP-100 remains approximately unchanged after removal of continuous radiation source", so I wonder is the luminescent photochromism reversible?
4. For better comparison between the structures of Th-SINAP-100 before and after irradiation, it would be helpful to list the unit cell dimensions and distances of pi-pi interaction as a table somewhere in the manuscript or supporting information.
5. The authors mentioned the high radioresistance of Th-SINAP-100 in the manuscript, so how is its chemical stabilities in other media, e.g. water and acid/basic solutions? This is also important for real-world applications.
6. What is the role of HCl during the synthesis of Th-SINAP-100? The yield of Th-SINAP-100 should be added in the manuscript.
7. Element analysis of Th-SINAP-100 should be provided.
8. FTIR spectrum of ligand should be included in Figure S8 for better comparison.
9. A few typos, Page 2 line 32 and line 35, "chromic". I believe the authors mean "color".
10. Some photochromic references relating donor-acceptor-type organic molecules are recommended. 1) Chem. Eur. J. 2019, 25, 13972–13976; 2) J. Mater. Chem. C, 2019, 7, 3100–3104; 3) J. Phys. Chem. C 2019, 123, 24670–24675.

REVIEWER COMMENTS

Reviewer #1 (Remarks to the Author):

The paper by Lin and co-workers provides an impressive demonstration of a dual-module photochromic metal-organic nanocluster, Th-SINAP-100, as a key element for colorimetric dosimetry for UV and ionizing radiations. The judicious synergy between radiation attenuators and photo responsive ligands is responsible for the rare color and PL simultaneously demonstrated in a single material.

While the methodology is well described and the results are very surprising especially taking into account the uncommon β -ray induced photochromic effect here observed, some aspects should be taken into account if continuous dose monitoring applications are envisaged here.

Response: The authors appreciated these positive comments.

The influence of the environmental conditions on the observed results should be assessed especially in terms of relative humidity and temperature.

Response: The influence of the environmental conditions including relative humidity and temperature was accessed by PXRD and TGA. PXRD studies further indicated that the structure of **Th-SINAP-100** remains intact at relative humidity (RH) of 35%, 55%, 75%, and 95% in air (Fig. S10a). In addition, PXRD and TGA demonstrated that **Th-SINAP-100** exhibits decent thermal stability and is stable up to 150 °C (Figs. S10c and S11).

Moreover, did the authors observe any recovery effect upon irradiation as function of time?

Response: We did not observe any recovery effect upon irradiation as a function of time. The coloration information can be stored and the PL intensity of **Th-SINAP-100** remains unchanged for at least 48 h even the irradiation is turned off (Fig. S7).

At which the extent the observed properties could be maintained at nanoscale, i.e how the radiation induced sensitivity could depend on the thickness on the sensitive material?

Response: To study the effect of thickness on the sensitivity to irradiation, **Th-SINAP-100** tablets with different thickness (0.15, 0.30, 0.60, and 0.90 mm) were prepared from polycrystalline samples. As shown in the following figure, the thicker material exhibits fast photoinduced response with

larger I_{532}/I_{432} ratio under identical irradiation conditions, suggesting that the radiation induced sensitivity is positively correlated with the thickness of the material.

The weak response observed at a low dose rate is very important for most of the relevant clinical applications. This is a shortcoming, could the author comment on potential degree of freedom to further extend the measuring of this intriguing nanocluster?

Response: One potential way to extend the measuring of **Th-SINAP-100** to low dosage range is to measure the quenching ratio of the PL band at 432 nm as a function of dose. As shown in Fig. 2d inset, the intensity of PL_{432} drastically decreased at low dose range. The quenching ratio $(I_0 - I)/I_0$ % (where I_0 and I are the PL intensities before and after irradiation) as a function of UV radiation dose was plotted in the figure below. Simple mathematical transformations give rise to linear correlation between $Dose/[(I_0 - I)/I_0]$ and UV dose. Similarly, such a method can also be used for ionization radiation dosimetry at low dose range.

Finally, demonstration of encapsulation of the sensitive nanocluster in simple shapes for envisioning potential applications (fibers, strips ...) is strongly encouraged.

Response: The authors appreciated this helpful suggestion. To facilitate the real-world application, the **Th-SINAP-100** polycrystalline can be further fabricated with polyvinylidene fluoride (PVDF) polymer into a flexible membrane (Fig. S8a), based on the method reported by Wang and coworkers (*Angew. Chem. Int. Ed.* 2020, 59, 11856). As shown in Fig. S8b, the color of membrane is initially purple under excitation with 254 nm UV light. Continuous UV irradiation (350 mA, 25 W) results in gradual transitions of emission colors from purple, to blue, to cyan, and to green, implying the potential applicability of **Th-SINAP-100@PVDF** membrane in radiation dosimetry.

Reviewer #2 (Remarks to the Author):

The evaluation of the crystallographic data reveals that the authors have alerts A in their cif check pdf. MOFs are porous structures, and therefore, it is expected to observe large voids. However, there are several methods to take them into account in the crystallographic models. Unfortunately, the authors do not address this problem properly. One way to treat the problem is to find and model the disordered solvent molecules which are located inside MOF pores. Instead, the authors have assigned the residual electron density to oxygen atoms which are randomly distributed in the voids of MOFs. Another way to treat this issue is to use the SQUEEZE procedure (freely available). However, in the case, the authors should count number of electrons and estimate the correct number of the solvent molecules inside a framework. Both these methods are commonly applied to the MOF structures.

To conclude, application of any of these methods is necessary for publication of the collected crystallographic data.

Response: The authors appreciated this helpful suggestion. SCXRD study revealed the presence of large electron density within the void space of **Th-SINAP-100**. The electron counts per cell for **Th-SINAP-100** were found to be 243.3 and 199.7 e^- based on *SQUEEZE* before and after γ -ray irradiation, respectively. These electron densities can be attributed to two types of solvent species including H_2O and DMF, making the estimation the number of solvent molecules rather challenging. Consequently, molecular formula of **Th-SINAP-100** was finalized as $[Th_6(OH)_4(O)_4(H_2O)_6](TPC)_8(HCOO)_4 \cdot xH_2O \cdot yDMF$.

Reviewer #3 (Remarks to the Author):

This manuscript presents a thorium nanocluster that is composed of $Th_6O_4(OH)_4$ cores decorated by photosensitive terpyridine derivatives. Interestingly, the as-reported compound Th-SINAP-100 exhibited a rather unusual dual-module photochromism upon multiple types of irradiations, including UV light, β -ray, and γ -ray. This feature allows visual broad dosimetry of ionization radiations over a range from 1 to 80 kGy. The mechanism of dual-module photochromism was

thoroughly studied by SCXRD and DFT calculations. This new colorimetric dosimeter based on nanocluster structure should be of great interest to a wide range of readers, including chemists and material scientists. I recommend this work to be accepted after addressing the following issues.

1. Molecular formula and compound name should be mentioned in the abstract.

Response: The molecular formula and compound name have been added in the abstract.

2. In Fig. 2a, rules in all images should be added.

Response: Scale bars have been added in Fig. 2a.

3. The author mentioned “The PL intensity of Th-SINAP-100 remains approximately unchanged after removal of continuous radiation source”, so I wonder is the luminescent photochromism reversible?

Response: The luminescent photochromism is reversible only at the second stage, of which the PL intensity of the emission at 532 nm decreases when Th-SINAP-100 is heated at 120 °C for 24 h. This reversible transition can be cycled for at least five times.

4. For better comparison between the structures of Th-SINAP-100 before and after irradiation, it would be helpful to list the unit cell dimensions and distances of pi-pi interaction as a table somewhere in the manuscript or supporting information.

Response: The unit cell dimensions have been listed in Table S1. The distances of π - π interaction have been added in Table S4.

5. The authors mentioned the high radioresistance of Th-SINAP-100 in the manuscript, so how is its chemical stabilities in other media, e.g. water and acid/basic solutions? This is also important for real-world applications.

Response: PXRD studies further indicated that **Th-SINAP-100** displays excellent chemical stability in different solutions with pH values ranging from 4 to 8 (Fig. S10b).

6. What is the role of HCl during the synthesis of Th-SINAP-100? The yield of Th-SINAP-100 should be added in the manuscript.

Response: The hydrochloric acid functions as a modulator for the synthesis of Th-SINAP-100. Synthesis without adding HCl resulted in amorphous precipitates rather than single crystals of Th-SINAP-100. The yield of Th-SINAP-100 was calculated to be 49(1)% based on Th. The related information has been added in the revised manuscript.

7. Element analysis of Th-SINAP-100 should be provided.

Response: SEM-EDS analysis of Th-SINAP-100 has been provided in Fig. S15 of the supporting information.

8. FTIR spectrum of ligand should be included in Figure S8 for better comparison.

Response: The FTIR spectrum of TPC ligand has been added in Figure S6.

9. A few typos, Page 2 line 32 and line 35, “chromic”. I believe the authors mean “color”.

Response: Fixed. And we also carefully checked the whole manuscript.

10. Some photochromic references relating donor-acceptor-type organic molecules are recommended. 1) Chem. Eur. J. 2019, 25, 13972–13976; 2) J. Mater. Chem. C, 2019, 7, 3100–3104; 3) J. Phys. Chem. C 2019, 123, 24670–24675.

Response: The recommended references have been cited in the revised manuscript.

REVIEWER COMMENTS

Reviewer #1 (Remarks to the Author):

The authors have well and convincingly addressed my concerns
The paper can be accepted for publication.

Reviewer #2 (Remarks to the Author):

The authors have addressed the observed Alert A using the recommended squeeze procedure. However, the check cif file still contains several alerts B which have not been addressed through the author's response in the cif file. I recommend doing so or re-considering the model which the authors applied for the crystallographic data analysis. In addition, the authors provide the explanation that "the electron densities can be attributed to two types of solvent species including H₂O and DMF, making the estimation the number of solvent molecules rather challenging." First, the authors should prove that the listed solvents are located inside the framework pores (e.g., through a digestion procedure). Second, the authors should estimate the amount of the solvent molecules through thermogravimetric analysis and compare with the number of solvent molecules predicted through the SQUEEZE procedure. This information should be included in the SI (both TGA analysis and digested samples accompanied with electron counting).

Reviewer #3 (Remarks to the Author):

The revised manuscript was improved considerably. But comment 7 "Element analysis of Th-SINAP-100 should be provided" should be re-answered again. As SEM-EDS is a semi-quantitative method for elemental analysis, this method is not available to reveal the accurate purity. Experiments of the contents for elements (such as C, H, N) are easy to carry out, and those are essential for the purity verification. After this revision, the submission can be accepted for publish.

REVIEWER COMMENTS

Reviewer #1 (Remarks to the Author):

The authors have well and convincingly addressed my concerns

The paper can be accepted for publication.

Response: We appreciated your evaluation of our manuscript.

Reviewer #2 (Remarks to the Author):

The authors have addressed the observed Alert A using the recommended squeeze procedure. However, the check cif file still contains several alerts B which have not been addressed through the author's response in the cif file. I recommend doing so or re-considering the model which the authors applied for the crystallographic data analysis. In addition, the authors provide the explanation that "the electron densities can be attributed to two types of solvent species including H₂O and DMF, making the estimation the number of solvent molecules rather challenging." First, the authors should prove that the listed solvents are located inside the framework pores (e.g., through a digestion procedure). Second, the authors should estimate the amount of the solvent molecules through thermogravimetric analysis and compare with the number of solvent molecules predicted through the SQUEEZE procedure. This information should be included in the SI (both TGA analysis and digested samples accompanied with electron counting).

Response: The author appreciated these helpful suggestions. Both Alerts A and Alerts B have been addressed in the CIF files and we have deposited the revised CIF files to CCDC. Elemental analyses of C, H and N were performed on Th-SINAP-100 with a Vario EL Cube elemental analyser. The weight contents of N, C, and H are 9.04%, 39.77%, and 3.775%, respectively, suggesting the presence of DMF and H₂O as solvent species in Th-SINAP-100. Based on the weight contents of N and C, the number of DMF molecules in the void of Th-SINAP-100 was calculated to be four per molecular formula. Furthermore, thermogravimetric analysis study reveals that an initial weight loss of 7.24w% occurring before 150 °C. Since the framework of Th-SINAP-100 remains intact at 150 °C as confirmed by PXRD study, this weight loss could be attributed to the departure of the solvent molecules (four DMF and one H₂O), in accordance with the calculated weight loss of 7.16w%. Furthermore, electron count per cell for Th-SINAP-100 was found to be 309 e⁻ per unit cell based on *SQUEEZE*, corresponding to 155 e⁻ per molecular formula, which is comparable with the electron count (170 e⁻) of four DMF and one H₂O molecules. As a consequence, the molecular formula of Th-SINAP-100 was determined to be [Th₆(OH)₄(O)₄(H₂O)₆](TPC)₈(HCOO)₄·4DMF·H₂O. The results of TGA and element analyses accompanied with electron counting have been included in the manuscript and SI.

Figure S11. TGA curve of Th-SINAP-100.

Reviewer #3 (Remarks to the Author):

The revised manuscript was improved considerably. But comment 7 "Element analysis of Th-SINAP-100 should be provided" should be re-answered again. As SEM-EDS is a semi-quantitative method for elemental analysis, this method is not available to reveal the accurate purity. Experiments of the contents for elements (such as C, H, N) are easy to carry out, and those are essential for the purity verification. After this revision, the submission can be accepted for publish.

Response: The author appreciated this helpful suggestion. The SEM-EDS study has been removed from the manuscript. Elemental analyses of C, H and N were performed on Th-SINAP-100 with a Vario EL Cube elemental analyser, showing that the weight contents of N, C, and H are 9.04%, 39.77%, and 3.775%, respectively. This result combined with the PXRD study confirms the purity of Th-SINAP-100.

REVIEWER COMMENTS

Reviewer #2 (Remarks to the Author):

The manuscript is suitable for publication.